# Pulmonary Function Tests Post-Stroke. Correlation between Lung Function, Severity of Stroke, and Improvement after Respiratory Muscle Training

Fotios Drakopanagiotakis [1] , Konstantinos Bonelis [1], Paschalis Steiropoulos [1] , Dimitrios Tsiptsios [2,*] ,
Anastasia Sousanidou [2] , Foteini Christidi [2], Aimilios Gkantzios [2] , Aspasia Serdari [3], Styliani Voutidou [2],
Chrysoula-Maria Takou [2], Christos Kokkotis [4], Nikolaos Aggelousis [4] and Konstantinos Vadikolias [2]

[1] Department of Respiratory Medicine, Medical School, Democritus University of Thrace,
University General Hospital of Alexandroupolis, 68100 Alexandroupolis, Greece;
fdrakopanagiotakis@gmail.com (F.D.); kon.bonelis@gmail.com (K.B.); steiropoulos@yahoo.com (P.S.)

[2] Department of Neurology, Medical School, Democritus University of Thrace,
University General Hospital of Alexandroupolis, 68100 Alexandroupolis, Greece;
anastasiasousanidou@gmail.com (A.S.); christidi.f.a@gmail.com (F.C.); aimilios.gk@gmail.com (A.G.);
stylvout@med.duth.gr (S.V.); chrytako1@med.duth.gr (C.-M.T.); vadikosm@yahoo.com (K.V.)

[3] Department of Child and Adolescent Psychiatry, School of Medicine, Democritus University of Thrace,
68100 Alexandroupolis, Greece; aserntar@med.duth.gr

[4] Department of Physical Education and Sport Science, Democritus University of Thrace, 69100 Komotini,
Greece; ckokkoti@affil.duth.gr (C.K.); nagelous@phyed.duth.gr (N.A.)

* Correspondence: tsiptsios.dimitrios@yahoo.gr

**Abstract:** Stroke is a significant cause of mortality and chronic morbidity caused by cardiovascular disease. Respiratory muscles can be affected in stroke survivors, leading to stroke complications, such as respiratory infections. Respiratory function can be assessed using pulmonary function tests (PFTs). Data regarding PFTs in stroke survivors are limited. We reviewed the correlation between PFTs and stroke severity or degree of disability. Furthermore, we reviewed the PFT change in stroke patients undergoing a respiratory muscle training program. We searched PubMed until September 2023 using inclusion and exclusion criteria in order to identify studies reporting PFTs post-stroke and their change after a respiratory muscle training program. Outcomes included lung function parameters ($FEV_1$, FVC, PEF, MIP and MEP) were measured in acute or chronic stroke survivors. We identified 22 studies of stroke patients, who had undergone PFTs and 24 randomised controlled trials in stroke patients having PFTs after respiratory muscle training. The number of patients included was limited and studies were characterised by great heterogeneity regarding the studied population and the applied intervention. In general, PFTs were significantly reduced compared to healthy controls and predicted normal values and associated with stroke severity. Furthermore, we found that respiratory muscle training was associated with significant improvement in various PFT parameters and functional stroke parameters. PFTs are associated with stroke severity and are improved after respiratory muscle training.

**Keywords:** pulmonary function tests; MIP; MEP; stroke; respiratory muscle training

## 1. Introduction

Stroke is one of the most important causes of death and disability. Symptoms are based on the location of the lesion and include minor to major motor and sensory deficits (including hemiplegia and gait disorders) [1]. Respiratory muscles, inspiratory and expiratory, may be affected and lead to changes in muscle distribution, muscle fiber architecture and strength generation. A respiratory muscle deficit, especially affecting the diaphragm, causes respiratory dysfunction affecting the proper contraction of the muscle [2]. Post-stroke lung changes (during the acute, subacute and chronic phases) have been documented. These

could include changed ventilatory patterns, increased aspiration risk, sleep difficulties and a high frequency of chest infections [3]. Given that they present a serious risk to patients, most of these clinical problems are recognised and handled in clinical settings [3–5].

Respiratory function can be evaluated by performing pulmonary function tests. Pulmonary function tests (PFTs) include spirometry, which assesses dynamic lung volumes such as forced expiratory volume in 1 s (FEV$_1$), forced vital capacity (FVC), peak expiratory flow (PEF) and static lung volumes and capacity (total lung capacity (TLC)) and residual volume (RV), performed most commonly by body plethysmography) [6]. The strength of respiratory muscles is assessed via maximum inspiratory pressure (MIP) and maximum expiratory pressure (MEP). PFTs are easily performed but cooperation of the participants is needed [6].

An impaired pulmonary function has been associated with increased incidence of cardiovascular diseases, including stroke, in several epidemiologic studies [7–9]. Duong et al. [10] performed a community-based cohort study, encompassing more than 126,000 patients, which showed that FEV$_1$ is an independent predictor of impairment and mortality of cardiovascular diseases, including stroke. Although PFTs have been examined as a possible risk factor marker of stroke, evidence is limited regarding the association of PFTs with the prognosis of patients, who have already suffered a stroke. Few studies have examined whether PFTs are a functional marker of stroke severity and prognosis [11–13]. PFTs, however, can provide useful information about the functional status of patients with stroke.

Rehabilitation programs for stroke survivors (depending on the motor defect) comprise a multidisciplinary approach to muscle exercise to improve the motion of the trunk, limbs and coordination of movement along gait [14]. Respiratory muscle training is an important part of these programs to improve respiratory function and reduce possible respiratory complications. Many studies have examined the effect of stroke on cardiopulmonary exercise capacity, measured either with treadmill or bicycle ergospirometers or walking tests after rehabilitation [14]. However, data are limited regarding the association of PFTs with functional improvement after a respiratory muscle training program [15].

The aim of this review is to examine whether PFTs could provide an index of stroke patient severity classification and follow-up. Moreover, we will review PFT changes post-stroke in patients undergoing respiratory muscle training.

## 2. Materials and Methods

We used the Preferred Reporting Items for Systematic Reviews and Meta-Analyses (PRISMA) checklist (CRD42023466356) as a guide for this study. The study's methods were a priori designed. We searched for studies including stroke survivors and performance of PFTs. These studies were further classified in studies examining baseline PFTs in stroke survivors and PFTs in stroke survivors who underwent a respiratory muscle training program. Therefore, the authors conducted literature research of two databases (MEDLINE and Scopus) for eligible studies. The key search terms were ((pulmonary function tests) AND (stroke)) OR ((pulmonary function tests) AND stroke AND (respiratory muscle training)). A search with the terms ((spirometry) AND (stroke)) OR ((spirometry) AND stroke AND (respiratory muscle training)) was also performed and did not retrieve additional studies. There were no sex restrictions and all articles that were published until September 2023 were retrieved. Observational, case-controlled studies and controlled trials were included. The title and abstract of the articles were screened by two different reviewers. Articles not related to the scope of this review (i.e., PFT measurement at baseline after stroke and how PFTs change after respiratory muscle training in patients with stroke) were removed. In total, 276 articles were retrieved for further analysis and their references were studied in order to search for other relevant studies. If the full-text article could not be retrieved, reference was included only if the abstract described all relevant information. After specific inclusion and exclusion criteria were used (Table 1), 46 articles were eligible for our review. Articles were eligible only if their results included measurements of at least one parameter of lung function tests in association to stroke patients. Primary parameters of interest included measurements of forced vital capacity (FVC), forced expiratory volume in 1 s

(FEV₁), peak expiratory flow (PEF), maximum expiratory pressure (MEP) and maximum inspiratory pressure (MEP) and how they are associated with stroke (Table 2).

**Table 1.** Inclusion and exclusion criteria.

| *Inclusion Criteria* | *Exclusion Criteria* |
|---|---|
| Abstracts written in English language<br>Patients with stroke<br>Adult patient population<br>PFTs performed post-stroke as baseline or RCTs after implementation of a respiratory muscle training program | Reviews, meta-analyses, editorials, case reports,<br>Articles in pediatric population<br>Articles in special populations (e.g., pregnancy)<br>Articles with outcomes related exclusively to exercise testing<br>PFTs performed in the setting of rehabilitation but not including a respiratory muscle training program |

In the second part of article retrieval, articles regarding correlation of PFTs with respiratory muscle training were reviewed. Only results of randomised controlled trials were included (Table 3). Data extraction was performed using a predefined data form created in Excel. Data regarding the author, year of publication, number of participants, the scale of stroke severity and prognosis, and the main results of each study were captured. We refrained from undertaking a meta-analysis or other statistical analysis, due to the high heterogeneity among the studies. Thus, the data were only descriptively analysed.

The screening and selection process is displayed in flow diagram 1 (Figure 1).

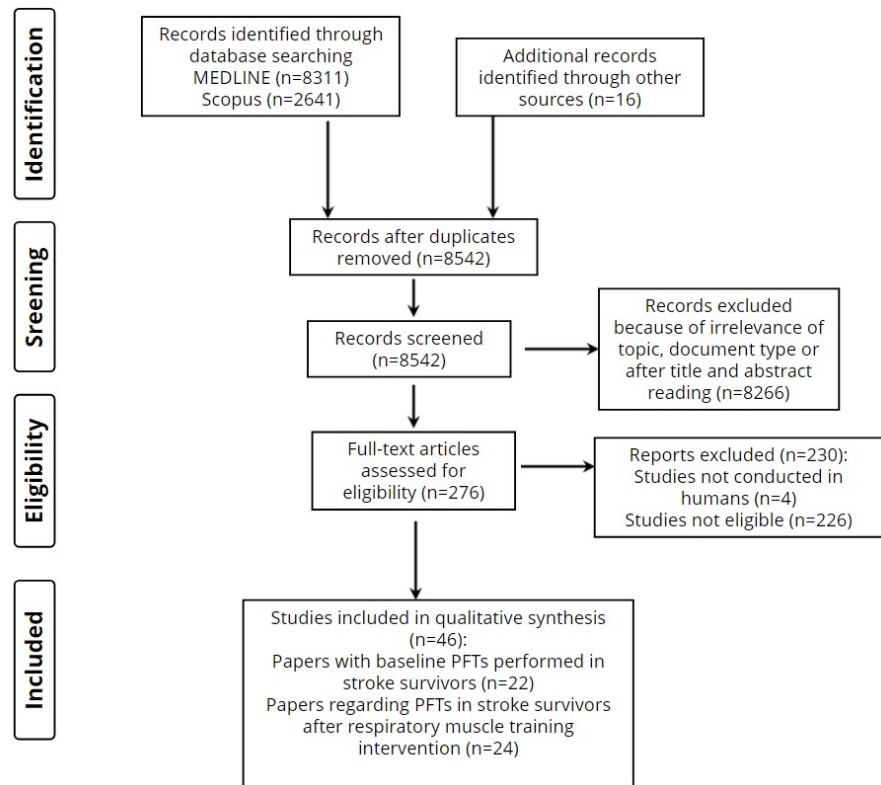

**Figure 1.** Screening and selection process.

### 3. Results

After the inclusion/exclusion criteria were evaluated, a total of 46 studies were included in this review. All the studies included stroke survivors and performance of PFTs. These studies were further classified in studies examining baseline PFTs in stroke survivors (n = 22) and PFTs in stroke survivors who underwent a respiratory muscle training program (n = 24). The main characteristics of the included studies are listed in Tables 2 and 3.

**Table 2.** Studies examining PFTs in post-stroke patients.

| Author/Date | Study Population/Mean Age/Gender (M/F) | Type of Study | Type of Stroke | Follow-Up Time | Scale of Stroke Severity | Study Aim | Results |
|---|---|---|---|---|---|---|---|
| Annoni J.M. et al. 1990 [11] | 23 non-smoking hemiplegic patients in the acute phase (53, 12/11) | Case–control | Any type | - | - | Correlation of PFTs with proximal arm function | FIVC and FEVC were reduced with time, independent of motor impairment but related to duration of illness. Patients exhibited a restrictive respiratory pattern. PEF and MEF were reduced by 75%. |
| De Almeida I.C. et al. [16] 2011 | 8 patients with right side hemiplegia ($51.25 \pm 13.8$, 4/4), 12 patients with left side hemiplegia ($55.33 \pm 9.57$, 4/8) and 8 controls ($52.12 \pm 7.28$, 5/3) | Case–control | Any type | - | Motor Assessment Scale | Comparison of PFTs and diaphragmatic excursion between groups | MIP significantly better in controls compared to patients with hemiplegia. $FEF_{25-75\%}$ and PEF significantly correlated to left diaphragmatic excursion. IC was not changed. No difference of FVC%, $FEV_1$%, $FEV_1$/FVC (small number of patients able to perform these PFT maneuvers) |
| Ezeugwu et al. 2013 [12] | 35 patients with stroke ($55.8 \pm 8.99$, 21/14) and 35 healthy controls ($55.6 \pm 9.03$, 21/14) | Case–control, cross-sectional | Any type | - | - | Comparison of PFTs between stroke patients and controls, correlation with chest excursion | Lower $FEV_1$, FVC and PEF in stroke patients. Obstructive and restrictive pattern in stroke patients. Lower chest excursion in stroke patients. No correlation between chest excursion and PFTs. |
| Fugl-Meyer et al. [17] 1983 | 54 patients with stroke and hemiplegia or hemiparesis | Cross-sectional | Any type | - | - | Correlation of PFTs with stroke severity | PFTs, MIP, MEP, lung compliance and resistance related to the degree of motor impairment and to the interval between stroke and investigation. Lower IC six months after stroke, more evident restrictive disturbance. |
| Jandt S.R. et al. 2011 [18] | 21 patients with stroke ($58.9 \pm 13.5$, 12/9) | Observative, descriptive | Any type | - | - | Correlation of PFTs with trunk impairment scale (TIS) | Significant correlation between TIS and PEF and between TIS and MEF. No correlation of TIS with $FEV_1$, FVC, $FEV_1$/FVC and MIF. |
| Jeong Y. et al. 2020 [13] | 52 patients with stroke within six months of onset (34/18) | Prospective | Any type | 4 weeks | NIHSS score, Berg Balance Scale | Correlation of PFTs at baseline and 4 weeks after rehabilitation with TIS, Berg Balance Scale and functional independence measure | Baseline FVC, $FEV_1$ and PEF correlated with initial TIS. Initial PEF significantly associated with Berg Balance Scale and Functional Independence Measure. No correlation with MIP and MEP. |
| Jung et al. 2014 [19] | 10 stroke patients ($59.7 \pm 12.9$, 8/2) and 16 healthy controls ($56.1 \pm 9.3$, 6/10) | Case–control | Any type | - | Korean Modified Barthel Index | Correlation of diaphragmatic excursion with PFTs | Restrictive PFTs in stroke patients. Left diaphragmatic excursion reduction correlated with reduced $FEV_1$ and FVC in stroke patients. |

**Table 2.** *Cont.*

| Author/Date | Study Population/Mean Age/Gender (M/F) | Type of Study | Type of Stroke | Follow-Up Time | Scale of Stroke Severity | Study Aim | Results |
|---|---|---|---|---|---|---|---|
| Khedr et al. 2000 [20] | 34 acute stroke patients (57.23 ± 13.26, 24/10) and 25 healthy volunteers (47.2 ± 22, 17/8) | Case–control, cross-sectional | Ischemic stroke | - | Scandinavian Stroke Scale | Comparison of diaphragmatic excursion and PFTs between groups; no PFTs in healthy controls | 41% of the stroke group had decreased diaphragmatic excursion and 70% decreased magnetic evoked potentials. Hemiplegic patients with restrictive PFTs. Negative correlation of $FEV_1$, FVC and $FEV_1/FVC$ with motor mobility and excitability threshold of affected hemisphere. |
| Kimura Y et al. 2013 [21] | 20 stroke patients without dysphagia (65.7 ± 8.1), 10 stroke patients with dysphagia (74.1 ± 10.2) and 10 healthy controls (68.2 ± 7.2) all male | Case–control, cross-sectional | Any type | - | Brunnstrom's recovery stage | Comparison of peak cough flow and spirometry between groups | Lower peak cough flow and IRV in stroke patients with dysphagia compared to healthy controls. Lower peak cough flow in stroke patients with dysphagia vs. without dysphagia. No differences in ERV or TV between groups. |
| Kulnik S.T. et al. 2016 [22] | 72 patients with stroke (64.6 ± 14.4, 42/30) | Single blind randomised control trial | Any type | 4 weeks | NIHSS score | Comparison of peak cough flow in voluntary and reflex cough | Weaker flow in patients' aspiration pneumonia. |
| Liaw M.Y. et al. 2016 [23] | 47 stroke patients with congestive heart failure (65.9 ± 11.5, 24/23) | Cohort | Any type | - | Brunnstrom stage, Barthel Index | Correlation of PFTs with Brunnstrom change | MIP negatively associated with Brunnstrom stage of the proximal and distal parts of the upper extremities and lower extremities, FVC, predicted FVC% and $FEV_1$%. MEP positively associated with average Brunnstrom stage of the distal area of the upper extremities, FVC, $FEV_1$, and $FEV_1/FVC$. $FEV_1/FVC$ negatively associated with the average Brunnstrom stage. Stroke patients had restrictive lung disorder and respiratory muscle weakness, associated with the neurological status of the affected limbs. |
| Lista Paz A. et al. 2016 [24] | 30 chronic stroke patients with a diagnosis of hemiplegia/hemiparesis who were able to walk (55.60 ± 15.84, 22/8) and 30 healthy controls (55.33 ± 14.61, 22/8) | Observational, cross-sectional | Any type | - | Scale Impact of Stroke version 16.0 | Comparison of MIP and MEP between groups | Significantly lower MIP and MEP in patients with stroke, <60%. Other spirometry parameters not measured. |

**Table 2.** *Cont.*

| Author/Date | Study Population/Mean Age/Gender (M/F) | Type of Study | Type of Stroke | Follow-Up Time | Scale of Stroke Severity | Study Aim | Results |
|---|---|---|---|---|---|---|---|
| Lista-Paz A. et al. 2023 [25] | 33 patients with stroke (56.9 ± 15.7, 24/9) and 33 healthy controls (56.2 ± 15.2, 24/9) | Observational, cross-sectional | Any type | - | Stroke Impact Scale version 16.0 | Comparison of PFTs and 6MWT between groups | Stroke patients had significantly lower lung volumes and capacities (VC, FVC, FEV$_1$, ERV, IC), than controls. Median FVC was 79% and PEF 64% of the reference value. The weak correlation of 6MWD with inspiratory reserve volume and PIF. |
| Luvizutto G.J. et al. 2017 [26] | 32 patients with acute stroke (14/18) | Cross-sectional | Ischemic stroke | - | NIHSS score, mRS score | Correlation of MIP and MEP with anthropometric data and neurologic severity | Lower MIP and MEP than predicted. No association with neurologic severity, positive association with BMI. Other spirometry parameters not measured. |
| Min S.W. et al. 2018 [27] | 57 patients with stroke (69.58 ± 10.29, 34/23) | Cross-sectional | Ischemic stroke | - | - | Correlation of PFTs with dysphagia and aspiration pneumonia | Increased dysphagia associated with worse PCF, FVC and FEV$_1$ values and aspiration pneumonia. |
| Nunez Filha M.C. et al. 2020 [28] | 53 patients with stroke (55 ± 13.43, 27/26) | Cross-sectional | Any type | - | NIHSS, Modified Barthel Index | Correlation of MIP and MEP and stroke severity with functional mobility | MIP, but not MEP, was independently associated with functional mobility in multivariate analysis. No other spirometry parameters were measured. |
| Pinheiro M.B. et al. 2014 [29] | 89 patients with stroke (56.2 ± 12.0, 48/41) | Cross-sectional, observational | Any type | 2 days | - | Correlation of MIP and MEP with stroke population (community vs. non-community ambulators) | Stroke subjects demonstrated decreases of 26.5 and 20% in the MIP and MEP. Significantly worse MIP values seen in non-community ambulators but not statistical significance of MEP, FEV$_1$ and FVC between community and non-community ambulators. |
| Santos R.S.D. et al. 2019 [30] | 44 patients with stroke (59.4 ± 12.2, 19/25) | Cross-sectional | Any type | - | Functional Independence Measure scale | Correlation of PFTs, MIP and MEP with TIS and Functional Independence Measure | Lower PFTs, MIP and MEP of predicted values, correlation of TIS with FVC, FEV$_1$ and MIP but not with MEP |
| Sezer N. et al. 2004 [31] | 20 patients with stroke (54.25 ± 11.42, 9/11) and 15 controls (9/6) | Cross-sectional | Any type | - | Brunnstrom classification stage, Barthel Index, Massachusetts General Hospital Functional Ambulation Classification | Comparison of cardiopulmonary response between groups | FEV$_1$, FVC, VC, PEF and MVV reduced in patients with stroke compared with controls but no correlation with motor disability. FEF$_{25-75\%}$ and FEV$_1$/FVC no different between groups. Significant respiratory dysfunction in hemiplegic patients. |

**Table 2.** *Cont.*

| Author/Date | Study Population/Mean Age/Gender (M/F) | Type of Study | Type of Stroke | Follow-Up Time | Scale of Stroke Severity | Study Aim | Results |
|---|---|---|---|---|---|---|---|
| Teixeira-Salmela L.F. et al. 2005 [32] | 16 community-dwelling stroke survivors (58.37 ± 15.47, 8/8) and 19 age-matched healthy subjects (60.21 ± 4.47, 9/10) | Descriptive case–control | Any type | - | - | Comparison of PFTs, MIP and MEP between stroke patients and controls | Significantly lower MIP and MEP in stroke patients compared to controls and decreased abdominal contribution to tidal volume. Dynamic lung volumes not measured. |
| Voyvoda et al. 2011 [33] | 23 hemiplegic patients (60.5 ± 10.7, 13/10) and 20 controls (61.2 ± 12.1, 13/7) | Descriptive case–control | Ischemic stroke | - | - | Comparison of diaphragm motility with ultrasonography and PFTs between groups | Significantly worse PFTs ($FEV_1$, FVC, $FEV_1$/FVC, MIP and MEP) in hemiplegic patients compared to control. No evidence of obstructive disturbance. No significance in diaphragmatic excursion between groups. |
| Xiao L.J. et al. 2020 [34] | 30 patients with stroke and dysphagia (53 ± 11, 20/10), 30 with stroke without dysphagia (59 ± 11, 17/13) and 30 healthy controls (55 ± 18, 18/12) | Descriptive case–control | Any type | - | - | Comparison PFTs between patients with dysphagia after stroke, patients without dysphagia and normal people. Correlation between swallowing function and pulmonary function. | Patients with dysphagia had significantly lower PEF, MIP, MEP FVC, $FEF_{25–75\%}$ and FIV but not $FEV_1$ compared to those without dysphagia. |

*3.1. Baseline PFTs in Stroke Survivors*

There was high variability in the number of participants included in the studies, as well of the characteristics of the examined patients. As shown in Table 2, only seven studies included more than fifty patients with stroke. Comparisons were made between the PFTs of stroke survivors with healthy controls and those with normal predicted values. The examined PFT parameters were not consistent among studies, particularly regarding the measurement of dynamic lung volumes (Table 2). Xiao et al. and Min et al. compared the PFTs of stroke patients with dysphagia and aspiration pneumonia with those without [27,34]. Stroke severity was assessed using various methods, including Trunk Impairment Score (TIS) [13,18,30], Berg Balance Scale [13] and Functional Independence Measure [30]. Diaphragmatic excursion and diaphragm dysfunction, assessed by ultrasound, were also measured in patients with stroke and correlated to PFTs [12,16,19,20,33].

*3.2. Lung Volumes and Flows in Stroke Patients*

Patients with stroke exhibited both an obstructive and a restrictive pattern of PFTs. The obstructive pattern was less often as the restrictive pattern and it was not associated with smoking or prior obstructive lung disease, since these were exclusion criteria in almost all the studies (Table 2). Specifically, Ezeugwu et al. compared the PFTs of 35 stroke survivors with 35 age-and sex-matched healthy controls [12]. They found that lung function, including $FEV_1$, FVC, $FEV_1/FVC$ and PEF was significantly worse in stroke patients. Obstructive pattern was the most common, seen in 46% of the patients, followed by a restrictive pattern in 38% of the stroke patients [12]. Fugl-Meyer at al. reported a restrictive pattern, associated with the degree of hemiplegia [17]. Stroke patients exhibited a restrictive PFT pattern in the study by Jung K. et al. with lower spirometric values compared to controls [19] and in the study by Kimura et al., in which stroke patients with and without dysphagia had lower VCs compared to healthy participants [21]. Lista-Paz et al. showed that the main lung volumes were significantly reduced in people with chronic stroke compared to healthy volunteers matched by age and sex (VC, IRV, IC, FVC, $FEV_1$ and PEF), as well as to their own reference values (VC, ERV, IC, FVC, $FEV_1$), results suggestive of a restrictive ventilatory defect in this population [25]. Sezer et al. also found a restrictive pattern with lower FVC and $FEV_1$ in stroke survivors [31]. Annoni et al. have also reported decreased VC in hemiplegic patients, more evident in patients with severe motor impairment and consistent with a restrictive pattern. The restrictive pattern could not be seen when forced dynamic volumes (FEVC and FIVC) were examined, but this was due to the small number of patients who were able to perform the respiratory maneuvers [11]. Khedr et al. and Liaw et al. also showed that stroke patients exhibited a restrictive PFT pattern [20,23]. Voyvoda et al. found a significantly lower FVC and $FEV_1$ in hemiplegic patients, suggestive of a restrictive pattern [33]. Liaw et al., who examined the PFTs of patients with stroke and congestive heart failure, found an FVC of $2.0 \pm 0.8$ L, with predicted FVC% of $67.9 \pm 18.8\%$, average $FEV_1$ of $1.6 \pm 0.7$ L, average predicted $FEV_1\%$ of $70.6 \pm 20.1\%$ and average $FEV_1/FVC$ of $84.2 \pm 10.5\%$, all lower than predicted and indicative of a restrictive pattern [23]. When patients with acute stroke were examined, values of FVC and $FEV_1$ were at 55% of those predicted and $FEV_1/FVC$ values were at 33% of those predicted [20]. In the study by Jandt et al., mean FVC and $FEV_1$ as well as PEF were all above 80% of those predicted, thus being in the limits of normal [18]. Only in a study of chronic stroke survivors living in the community, who were further classified according to their gait speed, were FVC and $FEV_1$ above 85% of those values predicted in the total population and $FEV_1/FVC$ was normal [29].

Expiratory flow has also been reported to be reduced in stroke patients: De Almeida et al. reported a significantly reduced PEF in patients with right hemiplegia [16] and so did Sezer et al. [31], Annoni et al. [11] and Ezeugwu et al. [12]. The peak cough flow was also reduced in the study carried out by Kimura et al. [21] and Lista-Paz et al. [25]. These results are supported by the study carried out by Kulnik et al., who showed that peak cough flow

was significantly reduced in patients with stroke [22] and in the study by Liaw et al., who showed a maximum mid-expiratory flow of $65.4 \pm 29.5\%$ [23].

### 3.3. MIP and MEP in Stroke Patients

Various studies measured inspiratory and expiratory pressure as the only PFT or in addition to other spirometric parameters. A common finding was the reduction in MIP and MEP in patients with stroke [16,23]. Lista Paz et al. reported decreased MIP and MEP in chronic stroke patients compared with healthy subjects. More importantly, values of MIP and MEP in stroke patients were also below 60% of those predicted in patients with stroke, which is the limit for defining muscle weakness [24]. In another study, the average MIP and MEP were $52.9 \pm 33.0$ cm $H_2O$ and $60.8 \pm 29.0$ cm $H_2O$, respectively, i.e., significantly low [23]. Similar results regarding MIP and MEP were reported from Luvizutto et al.: respiratory pressures were compared with the predicted value, a significant reduction in MIP was observed in the total sample and separately for men and women. When compared with the predicted values, a reduction in MEP was observed in the total sample, in men and in women [26]. MIP and MEP were reduced compared to normal values in the study by Pinheiro et al. [29] and Santos et al. [30].

### 3.4. Correlation of PFTs with Functional Impairment

Various scales were used in order to assess stroke severity: trunk impairment scale (TIS), the proximal arm function according to the British Medical Research Council, the Motor Assessment Scale, the Berg Balance Scale, the Functional Independence Measure, the Barthel Index, the Scandinavian Stroke Scale, Brunnstrom's recovery rate and function ambulation categories, the Motricity Index, the Scale Impact of Stroke, the NIHSS score, the functional reach test, the International Physical Activity Questionnaire, gait speed and functional ambulation classification (Table 2). Moreover, PFTs were associated in some studies with dysphagia scores or diaphragmatic excursion (Table 2).

Annoni et al. showed that VC was significantly associated with the severity of proximal arm function in hemiplegic patients and $FEV_1$ and FVC with the duration of stroke illness [11]. Santos et al. reported a significant correlation of $FEV_1$ and FVC with the TIS [30] and Jeong et al. reported that FVC and $FEV_1$ values correlated with the TIS scores at admission [13]. $FEV_1/FVC$ was negatively associated with the average Brunnstrom stage over the proximal and distal parts of the upper extremities and lower extremities and Barthel Index [23]. Jung et al. observed a significant positive correlation between left diaphragmatic excursion during deep breathing and $FEV_1$ (rho = 0.7, $p$ = 0.021) and FVC (rho = 0.86, $p$ = 0.007) in stroke patients. Diaphragmatic excursion did not correlate with the Korean Modified Barthel Index scores. There was found no correlation of these scores with PFTs in hemiplegic patients [19]. Sezer et al. also failed to report a significant correlation of PFTs with motor disability [31].

Abnormal magnetic potentials in the affected hemisphere and central conduction time have also been correlated to PFTs: a significant decline of disability score and higher excitability threshold percentage with lower $FEV_1$ were found in patients with reduced hemi-diaphragmatic excursion [20].

The expiratory flows PEF and MEF75% also correlated to functional impairment in the study by Annoni et al. [11]. Similar results were reported by Jandt et al. [18]. The latter showed that PEF was significantly associated with TIS. In the prospective study carried out by Jeong et al., the initial peak cough flow correlated with the TIS scores at admission [13]; the initial peak cough flow and FVC were predictive factors for the final TIS score. In linear regression analysis, the initial peak cough flow could predict test scores at discharge for the Berg Balance Scale and Functional Independence Measure [13]. The peak cough flow was also related to function ambulation categories in stroke patients with and without dysphagia [21].

Nunez Filha et al. showed that MIP was associated with stroke severity, as assessed with NIHSS [28] and Pinheiro et al. reported that MIP but not MEP had a negative correlation with gait velocity in stroke survivors [29]. Santos et al. reported a significant correlation of MIP with the TIS [30]. In patients with stroke and heart failure, MIP was negatively associated with the average Brunnstrom stage of the proximal and distal parts of the upper extremities and lower extremities [23].

MEP was positively associated with the average Brunnstrom stage of the distal area of the upper extremities [23]. Jandt et al. found that MEP and not MIP significantly correlated to TIS [18]. In acute stroke patients, however, a correlation of MIP and MEP with functional status could not be found [26]. As expected, a negative correlation of MIP and MEP with BMI was reported [26].

### 3.5. Correlation of PFTs with Dysphagia and Risk of Aspiration

Kimura Y. et al. compared the PFTs between stroke patients with and without dysphagia and healthy controls: peak cough flow, inspiratory reserve volume and VC were significantly lower in patients with stroke and dysphagia. Peak cough flow values significantly correlated to inspiratory reserve volume [21]. In a secondary analysis of trial data, Kulnik et al. found that increased peak voluntary cough flow and to a lesser extend peak reflex cough flow were associated with a lower possibility of aspiration pneumonia [22]. Min et al. reported similar results regarding peak cough flow, $FEV_1$ and FVC and their association with the dysphagia score [27]. Finally, Xiao et al. reported that patients with dysphagia had significantly lower PEF, MIP, MEP FVC, $FEF_{25-75\%}$ and FIV compared to those without dysphagia [34].

### 3.6. Correlation of PFTs with Diaphragmatic Dysfunction

De Almeida et al. measured the diaphragmatic excursion using ultrasound after the acute phase in stroke patients [16]. In their study, right-side hemiplegia affected the respiratory muscles more than left-side hemiplegia, as measured by MIP. Although spirometry was performed only in a few patients, $FEV_1$, PEF and $FEF_{25-75\%}$ were lower in patients with right-side hemiplegia. In right-side hemiplegia, movement was $4.97 \pm 0.78$ cm and $4.20 \pm 1.45$ cm for the right and left diaphragm, respectively, while in left-side hemiplegia, these values were $4.42 \pm 0.92$ cm and $4.66 \pm 1.17$ cm [16]. Jung et al. also evaluated the diaphragmatic motion using ultrasound, as mentioned earlier [19]: Stroke patients had a significant unilateral reduction in motion on the hemiplegic side [19]. Diaphragmatic excursion in patients with right-hemiplegia was lower than the one of controls on both sides [19]. On the contrary, in patients with left hemiplegia, diaphragmatic excursion was reduced only on the left side and increased on the right side. Left diaphragmatic motion during deep breathing correlated positively with FVC and $FEV_1$ [19]. Khedr et al. also reported that decreased hemi-diaphragmatic excursion was found in 41% of their patients, and it was associated with neurophysiological data of diaphragm, the degree of motor disability and respiratory dysfunction [20].

**Table 3.** Change in PFTs in stroke survivors undergoing a respiratory muscle training program.

| Author/Date | Study Population/MEAN Age/Gender (M/F) | Type of Study | Type of Stroke | Follow-Up Time | Scale of Stroke Severity | Study Aim | Results |
|---|---|---|---|---|---|---|---|
| Aydogan A.S. et al. 2022 [35] | 21 stroke patients: 11 in the treatment group (61.72 ± 10.77, 5/6) and 10 in the control group (66.10 ± 8.87, 2/8) | Single blinded randomised controlled trial | Any type | 6 weeks | - | PFTs, stroke severity scores before and after a neurodevelopmental treatment program and IMT in the treatment arm | Significantly better PEF and MIP in the treatment group |
| Britto R.R. et al. 2011 [36] | 18 patients with chronic stroke: 9 in the experimental group (56.66 ± 5.56, 5/4) and 9 in the control group (51.44 ± 15.98, 4/5) | Randomised controlled trial | Any type | 8 weeks | - | Comparison of MIP, inspiratory muscular resistance before and after IMT | Significantly better values for MIP and inspiratory muscular resistance in the intervention group compared to baseline |
| Chen P.C. et al. 2016 [37] | 21 patients with stroke and congestive heart failure: 11 in the IMT group (63.73 ± 14.64, 4/7) and 10 in the control group (67.50 ± 10.35, 4/6) | Randomised controlled trial | Any type | 10 weeks | Barthel Index | Comparison of spirometry, MIP and MEP between IMT group and control | Significant better values of $FEV_1$, FVC, MIP and Barthel Index in the intervention group compared to baseline and in MIP compared to the control group |
| Cho J.E. et al. 2018 [38] | 25 patients with stroke: 12 in the experimental group (47.58 ± 13.00, 7/5) and 13 in the control group (52.53 ± 9.06, 6/7) | Randomised controlled trial | Any type | 6 weeks | - | Comparison of diaphragm thickness ratio, MIP and inspiratory muscle endurance between IMT group and control | Increased diaphragm thickness, MIP and inspiratory muscle endurance in the IMP group |
| Guillen-Sola A. et al. 2017 [39] | 62 patients with dysphagia and stroke (69.0 ± 8.7, 38/24) | Randomised controlled trial | Ischemic stroke | 3 months | NIHSS score on admission, mRS score, Barthel Index on admission at Rehabilitation | Comparison of dysphagia score, MIP and MEP after a 3-week rehabilitation program and in 3 months between standard shallow therapy group, standard shallow therapy with IEMS and standard shallow therapy and neuromuscular electric simulation | MIP and MEP significantly improved in the standard shallow therapy with IEMS, compared to the other groups |

**Table 3.** *Cont.*

| Author/Date | Study Population/MEAN Age/Gender (M/F) | Type of Study | Type of Stroke | Follow-Up Time | Scale of Stroke Severity | Study Aim | Results |
|---|---|---|---|---|---|---|---|
| Jung K.M. et al. 2017 [40] | 12 patients with hemiparesis due to stroke: 6 in the experimental group ($61.2 \pm 4.2$, 2/4) and 6 in the control group ($62.2 \pm 5.3$, 3/3) | Randomised controlled trial | Any type | 4 weeks | - | Comparison of PFTs and walking ability between IMT group vs. aerobic exercise group | Significant improvement of $FEV_1$, FVC in both groups, significantly better $FEV_1$, FVC in the IMT group |
| Kilicoglou M.S. et al. 2022 [41] | 41 patients with stroke: 20 in the treatment group ($64.6 \pm 12.4$, 10/10) and 21 in the control group ($66.0 \pm 10.3$, 8/13) | Randomised-controlled trial | Any type | 6 weeks | - | Effect of respiratory exercise program on PFTs and diaphragm ultrasound parameters | FVC, $FEV_1$, $FEV_1$/FVC and diaphragm ultrasound parameters were improved after treatment in the intervention group |
| Kim C.Y. et al. 2015 [42] | 37 patients with post-stroke hemiplegia: 12 in the integrated training group ($57.53 \pm 7.73$, 7/5), 13 in the respiratory muscle training group ($59.20 \pm 6.12$, 6/7) and 12 in the control group ($60.53 \pm 0.38$, 4/8) | Randomised controlled trial | Any type | 6 weeks | - | Comparison of PFTs between controls, RMT and RMT plus abdominal drawing-in maneuver groups | Significantly better $FEV_1$, FVC and EMG diaphragm activation in the RMT and abdominal drawing-in maneuver group |
| Kim J. et al. 2014 [43] | 20 stroke patients: 10 in the exercise group ($54.10 \pm 11.69$) and 10 in the control group ($53.90 \pm 5.82$) | Randomised-controlled trial | Any type | 4 weeks | - | Effects of respiratory muscle and endurance training using an individualized training device for respiratory muscle training on PFTs and exercise capacity in stroke patients | FVC, $FEV_1$, PEF and 6MWT significantly better in the intervention group |
| Kulnik et al. 2015 [44] | 82 patients with stroke within two weeks of stroke onset ($64 \pm 14$, 49/33) | Single-blind randomized placebo-controlled trial | Any type | 4 weeks | NIHSS score | Change in peak expiratory cough flow in patients with IMT, EMT and no respiratory muscle training | Significantly better values compared to baseline in all groups with no effect of training |
| Lee K. et al. 2019 [45] | 25 chronic stroke patients, able to sit independently: 13 in the RMT group ($58.62 \pm 12.38$, 7/6) and 12 in the TSE group ($59.75 \pm 13.38$, 5/7) | Pilot randomised controlled trial | Any type | 6 weeks | mRS score | Comparison of PFTs between patients with progressive RMT with and without trunk stabilisation exercise | The MEP, PEF, MIP and PIF were significantly increased in the RMT group than in the control group |
| Lee D.K et al. 2018 [46] | 24 chronic stroke patients: 12 in the experimental group ($61.7 \pm 6.2$, 6/6) and 12 in the control group ($59.2 \pm 4.6$, 6/6) | Randomised controlled trials | Any type | 4 weeks | - | Comparison of PFTs, TIS and muscle activity of the trunk in patients who received neurodevelopmental treatment alone or with respiratory exercise | Significant better FVC, $FEV_1$, TIS, Rectus Abdominis, internal oblique and external oblique in the respiratory exercise group |
| Liaw M.Y. et al. 2020 [47] | 21 patients with stroke within six months of unilateral stroke, dysphagia, dysarthria and respiratory muscle weakness ($63.86 \pm 11.16$, 12/9) | Randomised controlled trial | Any type | 6 weeks | mRS score, Barthel Index | Comparison of PFTs after IERMT and control group | FVC, $FEV_1$ and MIP were significantly better in the intervention group |

**Table 3.** *Cont.*

| Author/Date | Study Population/MEAN Age/Gender (M/F) | Type of Study | Type of Stroke | Follow-Up Time | Scale of Stroke Severity | Study Aim | Results |
|---|---|---|---|---|---|---|---|
| Messagi-Sartor M. et al. 2015 [48] | 109 patients with subacute stroke (66.5 ± 11.2, 63/46) | Randomised controlled trial | Ischemic stroke | 6 months | NIHSS score, Barthel Index, mRS score | Comparison of MIP and MEP in the IEMT and the control group | Improved respiratory muscle strength in the intervention and control group. In IEMT group significantly improved MIP and MEP. Respiratory complications at 6 months more often in the control group, risk reduction of 14%. |
| Oh D. et al. 2016 [49] | 23 stroke patients: 11 in the experimental group (69.7 ± 6.8, 6/5) and 12 in the control group (71.6 ± 7.9, 7/5) | Randomised controlled trial | Any type | 6 weeks | - | Comparison of abdominal muscle thickness and PFTs of the IMT group vs. conventional therapy group | FVC, $FEV_1$, deep abdominal muscle thickness and Berg Balance Scale scores significantly improved in the experimental group |
| Parreiras de Menezes K.K. et al. 2019 [50] | 38 patients with stroke and respiratory muscle weakness: 19 in the experimental group (60 ± 14, 8/11), 19 in the control group (67 ± 11, 8/11) | Double-blind randomised trial | Any type | 8 weeks | - | Comparison of MIP, MEP, respiratory complications in the RMT group with high-intensity home-based program vs. control group | Significant increase in MIP, MEP, endurance of respiratory muscles and reduction of dyspnea in intervention group |
| Ptaszkowska et al. 2019 [51] | 60 stroke patients: 30 PNF-treated (64 ± 5, 20/10), 30 PNF-untreated (64 ± 7, 22/8) | Randomised controlled trial | Ischemic stroke | - | Barthel Index | Comparison of PFTs after respiratory stimulation through Proprioceptive Neuromuscular Facilitation (PNF) and controls | $FEV_1$/FVC% values in PNF-untreated group was substantially lower than in PNF-treated group |
| Rattes C. et al. 2018 [52] | 10 stroke patients with right hemiparesis (60 ± 5.7, 8/2) | Randomised controlled trial | Any type | 3 days | Barthel Index | Comparison of PFTs between respiratory stretching group and control | MIF, MEF and VT increased in respiratory stretching group compared to control group |
| Song G.B. et al. 2015 [53] | 40 patients with stroke: 20 in the CRE group (55.50 ± 11.43, 12/8) and 20 in the CEE group (58.30 ± 11.10, 11/9) | Randomised controlled trial | Any type | 8 weeks | - | Comparison of a chest resistance and a chest expansion intervention group regarding PFTs and TIS | Significantly better FVC, $FEV_1$ and TIS in both groups, TIS significantly better in the chest resistance intervention group |
| Sutbeyaz S.T. et al. 2010 [54] | 45 patients with stroke, randomised in three groups: 15 in IMT (62.8 ± 7.2, 8.7); 15 in breathing retraining, diaphragmatic breathing and pursed-lips breathing (60.8 ± 6.8, 8/7); 15 control group (61.9 ± 6.15, 8/7). | Randomised controlled trial | Any type | 6 weeks | Barthel Index | Effect of exercise--breathing retraining (BRT) and IMT--improve on cardiopulmonary functions | In IMT group significantly improved $FEV_1$, FVC, VC and $FEF_{25-75\%}$, compared with the BRT and control groups. PEF was increased significantly in the BTR group compared with the IMT and control groups. MIP and MEP increased in the BRT group and MIP in the IMT group compared with baseline and the control group. |
| Tovar-Alcaraz et al. 2021 [55] | 16 stroke survivors in the subacute phase: 8 in the experimental group (58 ± 12.9, 6/2) and 8 in the control group (56 ± 9.2, 6/2) | Randomised controlled trial | Any type | 8 weeks | Postural Scale for Stroke Patients (PASS), Berg scale | Comparison of MIP, PFTs, trunk and postural control in the IMT group vs. control | Significant increase in MIP compared to baseline in both groups, more significant in the IMT group |

**Table 3.** *Cont.*

| Author/Date | Study Population/MEAN Age/Gender (M/F) | Type of Study | Type of Stroke | Follow-Up Time | Scale of Stroke Severity | Study Aim | Results |
|---|---|---|---|---|---|---|---|
| Vaz L. et al. 2021 [56] | 50 patients with stroke with inspiratory muscle weakness (53 ± 11, 21/29) | Randomised controlled trial | Any type | 3 months | NIHSS score, Fugl-Meyer Assessment | Comparison of 6MWT, MIP, MEP in a group after IMT and without | Change in 6MWD in both groups but no difference in MIP, MEP after intervention |
| Yoo H.J et al. 2018 [57] | 40 patients with stroke: 20 in the intervention group (14/6) and 20 in the control group (12/8) | Randomised controlled trial | Any type | 3 weeks | NIHSS score, Modified Barthel Index, Berg Balance Scale, Fugl-Meyer Assessment | Comparison of PFTs and stroke severity scores in two groups assigned either to bedside IEMT or no intervention | PFTs significantly improved in the intervention group after 3 weeks of IEMT independent of the improvement in stroke-related disabilities |
| Zheng Y. et al. 2021 [58] | 60 patients within two months post-stroke: 30 in the experimental group (63.50 ± 10.36, 24/6) and 30 in the control group (67.23 ± 9.15, 19/11) | Randomised controlled trial | Any type | 3 weeks | Berg Balance Scale, Modified Barthel Index | Comparison of PFTs, stroke severity scores of the RMT group using Liuzijue Qigong vs. conventional respiratory training | Significant improvement in MIP, FVC and PEF in both groups, better MIP and MEP and TIS in the Liuzijue Qigong group |

*3.7. Change in PFTs after Respiratory Muscle Training in Stroke Survivors*

We further reviewed randomised controlled studies examining the effect of respiratory muscle training on PFTs in stroke survivors (Table 3). The number of patients included varied from 10–109. Only one study included more than 100 patients.

*3.8. Stroke Population*

The population of stroke survivors included in the studies was heterogeneous. Patients had suffered a subacute stroke within one to three weeks [39,48] and within two weeks [44] in the minority of studies, before respiratory muscle training started. Study participants who had suffered a stroke within six months of the start of respiratory muscle training were included in the studies carried out by Jung K.M. et al. [40], Kim J. et al. [43] and Tovar-Alcaraz et al. [55]. Zheng et al. [58] included patients with a stroke within two months and Yoo H.L. et al. [57] included patients undergoing respiratory muscle training after having a stroke within three months. Most of the studies examining the effect of respiratory muscle training included patients having suffered a stroke more than three months prior [35,38,43,45,47,49–53]. A few studies included patients up to five years after having suffered a stroke [50,56].

Inclusion criteria for the various studies were also heterogeneous: in various studies hemiparesis was an inclusion criterion [36,39–42,46,48,51,54–56]. Dysphagia was an inclusion criterion in the studies carried out by Guillen-Sola et al. [39], Liaw M.Y. et al. [47] and heart failure in the study by Chen P.C. et al. [37]. Age span, stroke functional scores and MIP scores differed among the studies. Most of the studies excluded patients with known pulmonary disease, with the exception of the studies carried out by Guillen-Sola et al. [39], Lee D.K. et al. [46] and Oh D. et al. [49].

*3.9. Type of Respiratory Muscle Training Intervention*

There was a high variability in interventions regarding the type of respiratory muscle training applied, the number of sets per session, the repetitions of each set, the duration of the program and the frequency of the training. The number of sets per session varied between one and ten sets and the repetitions in each set ranged between five and thirty (Table 3). One-time, short-effect interventions were also reported [52].

Nine studies reported inspiratory and expiratory muscle training in the intervention group [39,41,42,45–48,50,57], while one study compared inspiratory with expiratory muscle training [44] and another compared chest resistance with chest expansion training [53]. Most of the studies included performed both inspiratory with expiratory muscle training. Standard respiratory muscle training was compared to the Liuzijue training protocol in one study [58]. The programs were performed at home, at bedside or at hospital. Control groups also varied from the conventional stroke rehabilitation program to placebo respiratory training (Table 3). There were also studies, in which the standard respiratory muscle training involved the control group [58] or standard neuromuscular electrical simulation in patients with dysphagia [39].

As shown in Table 3, different follow-up schedules were used. PFTs of stroke survivors were measured after a course of respiratory muscle training program. The maximum respiratory muscle training duration was ten weeks [37]. A median value of 6 weeks of observation was performed in most of the studies. In the majority of the studies, the follow-up after the stroke regarding change in the pulmonary function tests coincided with the duration in respiratory muscle training. Only in three studies, the reported follow-up exceeded the duration of respiratory muscle training: Guillen-Solla et al. and Vaz L. et al. reported their results after a follow-up of three months [39,56], while in the study by Messagi-Sartor et al., the follow-up lasted six months [48].

*3.10. Outcomes Measured*

Spirometry parameters including $FEV_1$ and FVC were measured in the majority of studies [35,37,40–43,45–47,49,51,53–55,57,58]. The cardinal parameters reported were MIP and MEP. Scores of functional assessment of stroke severity, assessed by Trunk Impairment Score (TIS), Berg Balance Scale, Functional Independence Measure and diaphragm function were also reported.

IMT seems to increase PFTs, in particular MIP. Aydogan et al. reported no difference between groups regarding $FEV_1$, FVC, $FEV_1$/FVC, PEF, MIP and MEP prior to IMT [35]. After IMT, they reported a statistically significant increase in the intervention group regarding $FEV_1$ (0.30 ± 0.22 L), PEF (1,34 ± 1.22 L), MIP (14.9 ± 16.41 cm $H_2O$ and 15.76 ± 16.81% predicted) and MEP (13.54 ± 16.85 cm $H_2O$ and 7.73 ± 8.45% predicted) [35]. They reported a ca. 220 mL increase in FVC, which did not reach statistical significance. In the control group, only FVC reached statistical significance (0.32 ± 0.43 L) [35]. When the post-intervention values in the IMT group and controls were compared, only PEF and MIP were significantly better in the IMT group [35]. Britto et al. showed that MIP and inspiratory muscular endurance were significantly better in the IMT group compared to the control and that MIP improved significantly only in the intervention group [36]: MIP prior to intervention was 67.8 ± 14.6 cm $H_2O$ in the IMT group at baseline and rose to 102.2 ± 26.0 cm $H_2O$, while in the control group, MIP was 45.6 ± 13.8 cm $H_2O$ and increased to 56.7 ± 8.7 cm $H_2O$ [36]. In stroke patients with heart failure, a 10-week IMT program was associated with significantly better MIP, $FEV_1$, $FEV_1$% and FVC% compared to baseline [37]: IMT resulted in a 20.91 ± 19.73 cm $H_2O$ increase in MIP, a 0.22 ± 0.28 L increase in $FEV_1$ and 12.47 ± 13.52% predicted of $FEV_1$% predicted and 4.95 ± 6.75% predicted of the FVC. While no statistically significant changes were seen in the control group, a comparison between the IMT group and controls showed a significant increase in MIP in the IMT group [37]. Compared to aerobic exercise, an IMT program for 4 weeks was associated with a greater improvement in $FEV_1$, FVC and 6MWT, although better $FEV_1$ and FVC were seen in both groups [40]. A 6-week IMT program was also associated with significant improvement in MIP, inspiratory muscle endurance and diaphragm thickness [38]: the IMT program resulted in an impressive increase in MIP from 50.05 ± 21.92 cm $H_2O$ at baseline to 90.42 ± 30.91 cm $H_2O$ post IMT, while the increase in the control group was only 10 cm $H_2O$ [38].

Kilicoglu et al. reported a significant improvement in $FEV_1$ (0.15 ± 0.33 L or 7.30 ± 18.20% predicted) and FVC (0.31 ± 0.39 L or 2.65 ± 31.82% predicted) after respiratory training, which were associated with morphological parameters of diaphragm thickness, measured with ultrasound at baseline [41]. Similar results regarding $FEV_1$ and FVC were reported by Kim C.Y. et al. [42]. In the study by Oh et al., $FEV_1$ and FVC significantly increased by approximately 400 mL in the IMT group and the same applied for PEF (increase from 3.1 L/Min to 3.8 L/Min) [49]. Sutbeyaz et al. showed that IMT significantly improved spirometric values compared to a breathing retraining program or controls [54]: the mean increase in $FEV_1$ was 220 mL from 2.48 to 2.71 L compared to no difference in the other two groups and the mean increase in FVC was 230 mL, from 3.22 to 3.45 L [54]. Regarding pulmonary flows, no improvement was seen post-training in none of the groups while MIP and MEP increased in all three groups by approximately 2 to 7 cm $H_2O$ [54].

A combined IMT-EMT intervention in patients with stroke and dysphagia showed the greater improvement in MIP and MEP, while all groups had no differences regarding respiratory complications [39]: MIP increased by 21.1 ± 13.1 cm $H_2O$ at 3 weeks and by 18.3 ± 14.5 cm $H_2O$ in the three months after the IMT-EMT intervention, an increase of approximately 21% and 18%, respectively, while MEP increased by 26.4 ± 16.9 cm $H_2O$ in three weeks and 32.4 ± 21.2 cm $H_2O$ in three months (increase of 26.4% and 19.4%, respectively), almost two to three times greater as in the other two groups [39]. MIP was significantly improved by 45.90 ± 29.31 cm $H_2O$ after a combined respiratory muscle training program in patients with respiratory muscle weakness, dysphagia, and dysarthria compared to 5.45 ± 20.18 cm $H_2O$ in the control group [47]. Significant changes between

groups after the intervention were seen for $FEV_1$ and FVC [47]. FVC, $FEV_1$, PEF, 6MWT and Borg Dyspnea scores were significantly better after respiratory muscle training in the study by Kim J. et al. [43]. Improvements in $FEV_1$ and FVC after respiratory muscle training were related to better muscle activity of the trunk muscles [46,53]. In the study by Lee K. et al., the MIP, MEP, PEF, $FEV_1$, PIF and VC were significantly increased within the intervention and the control groups [45]. Regarding the between-group comparison, MIP, MEP, PEF and PIF were significantly increased in the RMT group compared with the control group [45]. MIP and MEP improvement after IEMT was also seen in patients with subacute stroke and was associated with a reduced rate of respiratory complications, including aspiration pneumonia [48]: MIP increased by 9.61 cm $H_2O$ or 10.2% more in the IEMT group compared to the controls, while MEP increased by 10.2 cm $H_20$ or 7% more. Home-based respiratory muscle training programs have been shown to achieve similar results regarding MIP and MEP [50]. An IMT of increasing intensity from 15% to 60% of MIP was associated with a greater improvement in MIP compared to the control group with a fixed load 7 cm $H_2O$ [55]; although there was no difference regarding $FEV_1$, FVC and PEF, MIP increased from $61.5 \pm 31.5$ to $80.5 \pm 35.1$ cm $H_2O$ in the intervention group [55]. Zheng et al. reported better PFTs and stroke functional scores after an intervention program using Liuzijue respiratory training compared to standard respiratory training [58].

Other studies have failed to show an additive beneficial effect of IMT or EMT in patients with hemiplegia, as measured using peak expiratory cough flow [44] or MIP and MEP [56] and improvements in pulmonary function after intervention were not always associated with improvements in the functional status post-stroke [57]. A single session of respiratory stimulation through Proprioceptive Neuromuscular Facilitation (PNF) showed an increase in $FEV_1/FVC$ but not of the other spirometric parameters, compared to the control group [51].

## 4. Discussion

Stroke represents the major cause of disability worldwide [59]. Stroke usually causes abnormalities in muscular tone, motor coordination and postural control [60]. Respiratory muscles can also be affected [61]. This leads to a decreased respiratory function due to respiratory muscle weakness. Besides the respiratory muscle insult, centrally induced changes in the respiratory efferent system can cause changes in the respiratory pattern and breathing frequency [61,62]. Moreover, since respiratory muscles do not function when isolated from the rest of the body, changes in the neck or trunk musculature due to spasticity and contracture in the hemiplegic site can lead to abnormal ventilatory patterns, with a dysfunction of the muscles of the affected side in cases of hemiparesis or hemiplegia and a compensatory hyperfunction of the muscles of the non-affected side, in order to ensure adequate minute ventilation [61]. These stroke-induced changes are mostly characterised by a restrictive pattern of lung function [15,61]. In extreme cases, the restrictive disturbance can be so severe that hypercapnia and hypoxemia may be seen, particularly in the acute and subacute phases [61,62].

Various studies have examined how respiratory muscle function, respiratory volumes, breathing rates, thoracic movement and cough efficacy are affected in stroke patients [16,33,63]. These changes are of particular interest in these patients since they can be used in order to assess the risk of a respiratory complication, such as aspiration or health-care-associated pneumonia or the risk of developing atelectasis [64].

Chronic dyspnea in patients after stroke is usually attributed either to the stroke per se, preexisting respiratory and cardiovascular conditions such as COPD or heart failure or to psychosocial reasons [65]. However, ventilatory changes are underdiagnosed [62]. This might lead to the initiation of treatments that are little effective (e.g., bronchodilators for patients with a restrictive pattern). Pulse oximetry and blood gas analysis are usually used to decide whether these patients need supplemental oxygen therapy.

PFTs are a simple, non-invasive, inexpensive method of assessing lung function and are widely used in respiratory medicine [6]. PFTs can also be performed at the bedside with simple equipment. Despite their obvious advantages, PFTs require the adequate cooperation of the examined subjects, in order to become valid results [6]. Due to aphasia, facial palsy, reduced consciousness, coordination disturbances or trunk muscle weakness, patients with stroke rarely undergo PFTs in order to assess their respiratory disorders [66,67]. More advanced techniques, such as cardiopulmonary exercise testing, are rarely used as diagnostic tools for the differential diagnosis of dyspnea in patients with stroke, since they are more time consuming and expensive and require excellent patient cooperation [15].

The aim of our review was to examine how PFTs change in post-stroke patients and whether respiratory muscle training can improve PFTs and thus respiratory function of these patients.

Results from the studies included in this review indicate that stroke patients have lower spirometric variables compared to healthy individuals and compared to normal reference values [12,17,19–21,23,25,31]. This involves mainly $FEV_1$ and FVC. The most common pattern observed was restrictive [12,17,19–21,23,25,31] while only one study reported an obstructive pattern, as suggested from the $FEV_1$/FVC ratio [12]. Other studies, however, have failed to show a reduction of forced dynamic lung volumes and only slow dynamic measurements such as VC were reduced in patients with stroke [11]. Interestingly, even if the above-mentioned values were normal, dynamic flow measurements such as PEF and $FEF_{25-75\%}$ were reduced, compared to healthy subjects [13,18]. PEF reduction is important, since it depicts a reduced cough reflex. More consistent results were seen when the inspiratory and expiratory pressures of MIP and MEP were examined, suggesting that both inspiration and expiration are affected [16,24,26,29,30].

Reductions in PFTs are associated with reduced functional status and reduced trunk control in patients with stroke in our review of the literature. $FEV_1$, FVC, MIP, MEP and PEF were associated with TIS, Barthel Index or Brunnstrom score [11,13,18,20,21,28–30]. These results are in accordance with a recently published meta-analysis of trunk control ability and respiratory function in stroke patients [68]. PFTs are also associated with diaphragm excursion in studies using ultrasonography [16,19,20]. However, there is not a single PFT value, which is consistently reduced in patients with stroke, therefore making standardisation of pulmonary functional impairment difficult. The lack of a single PFT value with a 100% sensitivity and specificity for the diagnosis of respiratory disease is not uncommon: even for extremely common pulmonary diseases, such as asthma and COPD, spirometry should be interpreted as a whole and many times repeated at different timepoints, in order to set a valid diagnosis [10,67,69]. The same principle seems to apply for patients with stroke.

It is important to notice that not only PFTs can be affected due to stroke but also that impaired lung function parameters are a risk factor for the development of cardiovascular complications, including stroke [10]. This is shown in many epidemiological studies, examining the predictive value of $FEV_1$ and FVC [10,69–71]. One could assume that reduced $FEV_1$ or FVC might represent underdiagnosed cases of COPD, which is known to affect the cardiovascular system, mainly due to the common detrimental effects of smoking in systemic inflammation [69]. However, this is not the only explanation, since restrictive PFTs (therefore exclusive of COPD) are also associated with increased cardiovascular risk [70]. In addition, COPD is clearly associated with an increased risk of hemorrhagic strokes, but it is unclear whether the increased frequency of ischemic stroke in COPD is the result of COPD per se or of a confounding effect [72]. Despite the presence or not of an etiologic correlation, the fact is that COPD patients who suffer a stroke have worse prognosis than patients with stroke without COPD [73].

On the other hand, the association of stroke severity indexes with reduced PFTs can increase our awareness of possible respiratory complications, thus enhancing a more proactive respiratory management of these patients. Therefore, we reviewed whether the respiratory function of patients with stroke can be improved, particularly with respiratory

muscle training. PFTs showed a significant improvement after respiratory muscle training in the majority of the studies, particularly regarding MIP and MEP, although other PFT parameters were also increased [35–38,41,42,45,47,54]. Pai and Li reported that only MIP and MEP were significantly better after the respiratory muscle training and that the lack of improvement seen in $FEV_1$ and FVC might be the result of a too-short training period [68]. In their meta-analysis, Pozuelo-Carascosa et al. reported that respiratory muscle training improved $FEV_1$, FVC, PEF, MEP, MIP and walking ability assessed using the 6MWT [74]. The RMT interventions were associated with a 12.2% increase in $FEV_1$% predicted while FVC improved by 6.75% [74] PEF also increased by 46.97 L/sec and MEP and MIP improved their baseline values by 10.05 and 22.40 cm $H_20$, respectively [74]. Other meta-analyses have reported similar findings [2,75]. However, results are not consistent; Fabero-Garrido et al. reported in their meta-analysis that only MIP and PEF were affected from respiratory muscle training in the short term but not $FEV_1$ and MEP, although the difference in $FEV_1$ was close to statistical significance [15].

We did not examine changes in our review regarding walking capacity or exercise capacity using ergospirometry. Pozuelo-Carascosa et al. reported that the walking distance increased, although minimally [74]. However, small differences can have significant implications in patients' daily activities. Despite the PFTs improvement dyspnea, the Barthel Index and Berg Balance Scale were not improved [74]. Lack of improvement regarding quality of life and functional improvement has been reported in other studies [36,56].

The reasons for the lack of consistent changes in PFTs throughout the studies are not always clear. The small number of participants and the different protocols used make a results comparison difficult. However, it is important to keep in mind that the evaluation of pulmonary function directly after rehabilitation might not be showing all benefits: duration of follow-up lasted in most of the studies three to eight weeks and pulmonary function tests were reported directly at the end of the rehabilitation program. Only three studies in our review reported follow-up of three to six months after a rehabilitation program [39,48,56].

Both IMT and combined IMT and EMT showed positive results, although attempts have been undertaken to further improve the respiratory muscle training [58]. This is in accordance with the findings of Pozuelo-Carrascosa et al. [74]. The effects of respiratory muscle training were in addition to the physiotherapy program, which includes components known as physiotherapy, strength training and aerobic exercise.

The most important clinical consequence of the PFTs improvement after respiratory muscle training is the reduction of respiratory complications, as shown in the meta-analysis of Menezes et al. [76] analysing the results of the studies by Kulnik and Messagi-Sartor et al. [44,48]. Therefore, bedside and home-based care to improve trunk control and respiratory muscles can improve respiratory function.

## 5. Limitations

Several limitations apply when interpreting PFTs in stroke patients; apart from patient cooperation, studies recruited a small number of participants, in different stroke populations, with different intervention protocols and different measured outcomes. As mentioned in the discussion, measurements of PFTs were mostly carried out immediately at the end of the respiratory muscle training program, thus missing beneficial effects of training which could occur later in the course of recovery. Very few studies have reported consecutive PFTs in stroke patients. Thus, a baseline snapshot and a follow-up measurement in short time intervals cannot lead to safe conclusions regarding the long-term consequences of stroke regarding the respiratory function and the duration of improvement after a rehabilitation program. Moreover, to avoid bias, almost all studies have excluded patients with pulmonary comorbidities prior to stroke. However, this excludes the extrapolation of results in the majority of the patients in clinical practice who have concomitant cardiovascular and pulmonary risk factors.

## 6. Conclusions

Considering the results of the studies reviewed, it seems that PFTs are affected after stroke and this is seen in stroke survivors in the subacute and chronic phase. Although not performed routinely in stroke patients, PFTs can provide valuable information regarding the risk of further respiratory complications and correlate to stroke severity scores. Although, there is not a single PFT marker, which is reduced in all patients, MIP and MEP might be more sensitive to identify patients at risk. Respiratory muscle training can significantly improve PFTs, thus reducing respiratory complications. However, the small number of studies and study participants, with differences in the study protocols, underlines the need for further research in this field.

**Author Contributions:** F.D. and K.B. reviewed the literature, screened the abstracts of the reference list, deleted duplicates and citations not meeting the inclusion criteria and assessed the articles; P.S. and A.S. (Anastasia Sousanidou) solved any disagreements regarding screening or the selection process; F.D. and K.B. wrote the manuscript; A.G., P.S., F.C., C.K., A.S. (Aspasia Serdari) and D.T. reviewed the tables, the presentation of the data and the methodology; P.S., S.V., N.A. and K.V. carried out the review and editing; F.D., K.B. and C.-M.T. wrote the final version. All authors have read and agreed to the published version of the manuscript.

**Funding:** We acknowledge the support of this work by the project "Study of the interrelationships between neuroimaging, neurophysiological and biomechanical biomarkers in stroke rehabilitation (NEURO-BIO-MECH in stroke rehab)" (MIS 5047286), which was implemented under the action "Support for Regional Excellence" funded by the operational program "Competitiveness, Entrepreneurship and Innovation" (NSRFm2014-2020) and co-financed by Greece and the European Union (the European Regional Development Fund).

**Institutional Review Board Statement:** Not applicable.

**Informed Consent Statement:** Not applicable.

**Data Availability Statement:** All data discussed within this manuscript are available on PubMed.

**Conflicts of Interest:** The authors declare no conflicts of interest.

## Abbreviations

BRT: exercise-breathing retraining, EMG: electromyography, EMT: expiratory muscle training, ERV: expiratory residual volume, $FEF_{25-75\%}$: forced expiratory flow 25–75, $FEV_1$: forced expiratory volume in 1 s, FEVC: forced expiratory vital capacity, FIVC: forced inspiratory vital capacity, FVC: forced vital capacity, IC: inspiratory capacity, IMT: inspiratory muscle training, MEF: maximal expiratory flow, MEP: maximal expiratory pressure, MIP: maximal inspiratory pressure, NIHSS: National Institutes of Health Stroke Scale, PEF: peak expiratory flow, PNF: Proprioceptive Neuromuscular Facilitation, PFTs: pulmonary function tests, RCT: randomised controlled trial, TIS: trunk impairment scale, VC: vital capacity

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
