# Peer review of "Pulmonary Function Tests Post-Stroke. Correlation between Lung Function, Severity of Stroke, and Improvement after Respiratory Muscle Training"

_2035-8377, doi:10.3390/neurolint16010009_

Round 1
Reviewer 1 Report
Comments and Suggestions for Authors
Dear authors,
Thank you for summarising the studies on pulmonary training in stroke patients. I miss the information in the Result part on how long time follow-ups were reported in the studies. One study reported 6 months follow-up and another 3 months follow-up according to Table 3, while the remaining were reporting weeks. Please, include information on that in the article (Results) and mention in the Discussion. Evaluation directly after rehabilitation maybe is not showing all benefits. Connecting to that, you should mention how long time in a total the programs/trainings lasted. Probably they correspond to the time of follow-up in Table 3. Please, develop more that issue which is important from the clinical point of view.
Comments on the Quality of English Language
There som misspellings, for ex line 425.
Author Response
Dear Reviewer,
Many thanks for your time spent reviewing our manuscript.
Appropriate modifications were made to the text as follows.
Thank you for summarising the studies on pulmonary training in stroke patients. I miss the information in the Result part on how long time follow-ups were reported in the studies. One study reported 6 months follow-up and another 3 months follow-up according to Table 3, while the remaining were reporting weeks. Please, include information on that in the article (Results) and mention in the Discussion. Evaluation directly after rehabilitation maybe is not showing all benefits. Connecting to that, you should mention how long time in a total the programs/trainings lasted. Probably they correspond to the time of follow-up in Table 3. Please, develop more that issue which is important from the clinical point of view.
- Thank you for your important comment. Indeed, the vast majority of the studies reported a follow-up equal to the duration of respiratory muscle training. We have included this information in the Results, Discussion and Limitations session as proposed.
There some misspellings, for ex line 425.
- Thank you for your comment. We have gone once more through the document, correcting misspellings.
Looking forward to your follow up comments.
Yours Sincerely,
Dr Tsiptsios
Reviewer 2 Report
Comments and Suggestions for Authors
In this systematic review, Drakopanagiotakis et al. summarized the literature on correlation between lung function, severity of stroke, and improvement after respiratory muscle training. The topic is interesting, since the consequences of low lung functions can be extensive. I have the following comments:
- The objective of this review was postulated since there are only few studies investigating the effect of respiratory muscle training after stroke. However, there is another systematic review and meta-analysis published two years ago and also mentioned in the introduction. What is the difference to this previous review? Relevant new studies on this topic? Other inclusion criteria?
- Did the authors potentially miss some articles since no additional spelling/wording was included in the searching string? E.g. spirometry instead of pulmonary function test.
- Results 3.1 is less presented as results than as methods.
- Is there some literature if the lung functions are already impaired before the stroke since stroke and COPD share common risk factors?
- Results 3.2: The authors provide lung function parameters that are altered to controls, however, are also some of the lung function parameters not altered?
- Results are a bit confusing since most of the studies investigated a unique correlation and therefore are not comparable among each other. However, perhaps there is a way of presenting this a bit more clearly?
- Results 3.8: I suggest that the authors wanted to mention the time the stroke happened before respiratory muscle training started?
Author Response
Dear Reviewer,Many thanks for your time spent reviewing our manuscript.
Appropriate modifications were made to the text as follows.
- The objective of this review was postulated since there are only few studies investigating the effect of respiratory muscle training after stroke. However, there is another systematic review and meta-analysis published two years ago and also mentioned in the introduction. What is the difference to this previous review? Relevant new studies on this topic? Other inclusion criteria?
Thank you for this important comment. In the systematic review and meta-analysis of Fabero-Garrido et al., the authors included nine studies. In our review, we included twenty-four studies. Four studies were published in 2021 and 2022 (Aydogan A.S. et al, Kilicoglou M.S. et al.,Tovar-Alcaraz et al. and Vaz L. et al.), therefore they were probably not included in the study of Fabero-Garrido et al. The inclusion and exclusion criteria were similar between the two systematic reviews. However, in the systematic review of Fabero-Garrido et al. the authors mostly included studies examining the effect of respiratory muscle training on exercise testing.
- Did the authors potentially miss some articles since no additional spelling/wording was included in the searching string? E.g. spirometry instead of pulmonary function test.
Thank you very much for this valuable comment. We did perform our search using ‘spirometry’ as a searching term. No new studies were identified compared to ‘pulmonary function tests’ as a search term. We included ‘spirometry’ as a search term in the Methods of the revised manuscript.
- Results 3.1 is less presented as results than as methods.
Thank you very much for your comment. We have rewritten Results 3.1, according to your suggestion.
- Is there some literature if the lung functions are already impaired before the stroke since stroke and COPD share common risk factors?
Thank you very much for your important comment. Reduced lung function parameters have been associated with increased risk of cardiovascular morbidities, including stroke. An explanation might be, that subjects with reduced FEV1 might present undiagnosed COPD patients. With the data available, it is unclear whether COPD is directly associated with increased risk of ischemic stroke or if it acts as a confounding factor. As you mentioned, due to the common risk factors, an etiologic relation sounds plausible. We address this issue in the discussion.
- Results 3.2: The authors provide lung function parameters that are altered to controls, however, are also some of the lung function parameters not altered?
Thank you very much for this astute observation. We have added negative results and further expanded positive results in table 2. After carefully examining the results of the studies included, we did not find pulmonary function parameters that were consistently not altered.
- Results are a bit confusing since most of the studies investigated a unique correlation and therefore are not comparable among each other. However, perhaps there is a way of presenting this a bit more clearly?
Thank you very much for this comment. As you have pointed out in your thoughtful review, the limited comparability of the studies and the investigation of unique correlations reduces the robustness of our findings. Therefore, we chose to go with a more descriptive way of presenting the results, avoiding any overstatement of the findings. Moreover, to improve clarity, we have rewritten the results in paragraphs 3.2, 3.3, 3.4 and 3.10.
- Results 3.8: I suggest that the authors wanted to mention the time the stroke happened before respiratory muscle training started?
Thank you very much for this observation. We have adapted Results 3.8 regarding respiratory muscle training according to your suggestion.
Looking forward to your follow up comments.
Yours Sincerely,
Dr Tsiptsios
Reviewer 3 Report
Comments and Suggestions for Authors
The use of 2 databases is hardly comprehensive.
I would much prefer specific results. They are often not provided.
Presentation of data in Table 3 is poorly arranged.
Was a meta-analysis not possible with some of the data?
Functional Independence Measure is a formal name.
References are presented inconsistently.
Comments on the Quality of English LanguageNot a major problem
Author Response
Dear Reviewer,
Many thanks for your time spent reviewing our manuscript.
Appropriate modifications were made to the text as follows.
The use of 2 databases is hardly comprehensive.
Thank you for your important comment. We have partially rewritten Section 2.Materials and Methods and included the sentence: ‘We searched for studies including stroke survivors and performance of PFTs. These studies were further classified in studies examining baseline PFTs in stroke survivors and PFTs in stroke survivors who underwent a respiratory muscle training program’.
I would much prefer specific results. They are often not provided.
Thank you very much for this valuable observation. We have extensively rewritten the results, particularly Sections 3.2, 3.3, 3.4 and 3.10 providing more specific results of the studies included.
Presentation of data in Table 3 is poorly arranged.
Thank you for this observation. Data in table 3 have been better arranged and the layout has been changed to portrait.
Was a meta-analysis not possible with some of the data?
Thank you very much for your valuable comment. We refrained from conducting a meta-analysis, since meta-analyses have been published and are cited in the manuscript and due to the high heterogeneity among studies. We present the results of 24 studies regarding the effect of respiratory muscle training (more than those included in published meta-analyses), preferring to present the great variability of indication, duration, timing and protocols applied.
Functional Independence Measure is a formal name.
Thank you very much for this observation. The term has been corrected according to your remark.
References are presented inconsistently.
Thank you very much for this remark. We have checked the manuscript and added relevant references, where missing.
Looking forward to your follow up comments.
Yours Sincerely,
Dr Tsiptsios
Round 2
Reviewer 2 Report
Comments and Suggestions for Authors
My comments were addressed and implemented into the revised version of the manuscript. I have no additional comments.